# Nano-Sized Extracellular Vesicles Secreted from GATA-4 Modified Mesenchymal Stem Cells Promote Angiogenesis by Delivering Let-7 miRNAs

**DOI:** 10.3390/cells11091573

**Published:** 2022-05-07

**Authors:** Min Gong, Min Wang, Jie Xu, Bin Yu, Yi-Gang Wang, Min Liu, Muhammad Ashraf, Meifeng Xu

**Affiliations:** 1Department of Pathology and Laboratory Medicine, University of Cincinnati Medical Center, Cincinnati, OH 45267, USA; gongmin0794@suda.edu.cn (M.G.); wang.min1@mayo.edu (M.W.); xuje@ucmail.uc.edu (J.X.); binyu76@gmail.com (B.Y.); wanyy@ucmail.uc.edu (Y.-G.W.); lium@ucmail.uc.edu (M.L.); 2Department of Neonatology, Children’s Hospital of Soochow University, No. 92 Zhongnan Street, Suzhou 215025, China; 3Department of Cardiovascular Diseases, Mayo Clinic, Rochester, MN 55905, USA; 4Department of Medicine, Cardiology, Medical College of Georgia, Augusta University, Augusta, GA 30912, USA; mashraf@augusta.edu

**Keywords:** extracellular vesicles, mesenchymal stem cells, GATA-4, angiogenesis, let-7

## Abstract

We demonstrated previously that extracellular vesicles (EVs) released from mesenchymal stem cells (MSCs) play a critical role in angiogenesis. Here, we examine whether this pro-angiogenic efficacy is enhanced in EVs derived from MSCs overexpressing GATA-4 (MSC^GATA−4^). Methods and Results. EVs were isolated from MSC^GATA-4^ (EV^GATA-4^) and control MSCs transduced with an empty vector (EV^null^). EVs from both cell types were of the same size and displayed similar molecular markers. Compared with EV^null^, EV^GATA-4^ increased both a tube-like structure formation and spheroid-based sprouting of human umbilical vein endothelial cells (HUVECs). The EV^GATA-4^ increased the numbers of CD31-positive cells and hemoglobin content inside Matrigel plugs subcutaneously transplanted into mice for 2 weeks. Moreover, EV^GATA-4^ encapsulated higher levels of let-7 family miRs compared to EV^null^. The transfer of exosomal let-7 miRs into HUVECs was recorded with an accompanied down-regulation of thrombospondin-1 (THBS1) expression, a major endogenous angiogenesis inhibitor. The loss-and-gain of function studies of let-7 miRs showed that let-7f knockdown significantly decreased EV^GATA-4^-mediated vascularization inside Matrigel plugs. In contrast, let-7f overexpression promoted HUVEC migration and tube formation. Conclusion. Our results indicate that EVs derived from genetically modified MSCs with GATA-4 overexpression had increased pro-angiogenic capacity due to the delivery of let-7 miRs that targeted THBS1 in endothelial cells.

## 1. Introduction

Therapeutic angiogenesis has been recognized as a very important strategy for treating ischemic diseases such as peripheral and coronary vascular disease, cerebral infarction, and critical limb ischemia. Mesenchymal stem cells (MSCs) are widely used in research and pre-clinical tests to treat ischemic diseases [1,2]. Indeed, results obtained from both in vitro and in vivo experiments confirm that MSCs can enhance new blood vessel formation [2,3,4]. Many studies, including ours, show that MSCs secrete paracrine factors, including extracellular vesicles (EVs), that orchestrate interactions within the cellular milieu to promote angiogenesis [3,5,6,7,8,9,10]. EVs are characterized as exosomes, microvesicles, and apoptotic bodies, according to the nomenclature established by the International Society for Extracellular Vesicles [11]. EVs, especially exosomes, are considered to play a critical role in cardiac regeneration and protection [12,13,14,15] since they can recapitulate the regenerative and functional effects produced by their parent cells [9,16,17].

EVs are bi-lipid-layered vesicles containing proteins, mRNAs, and microRNAs (miRs) and can transfer their contents to the recipient cells to modulate the function of these cells [18,19,20]. miRs are small, evolutionarily conserved, well-characterized endogenous noncoding RNA molecules that can negatively regulate gene expression by targeting specific mRNAs, leading to degradation or translational repression [21]. Recent reports indicate that several miRs can promote angiogenesis [22,23,24,25]. The content of EVs can be altered by preconditioning parent cells. Pro-angiogenic exosomal miRs increase substantially following hypoxic/ischemic preconditioning [26,27]. More importantly, EVs derived from preconditioned stem cells have a high potential for increasing vascular density, reducing cardiomyocyte (CM) apoptosis and myocardial fibrosis [27], and increasing recruitment of cardiac progenitor cells to the infarcted heart [28]. EVs collected from stem cells transfected with specific miRs produce greater beneficial effects in protecting CM and regenerating infarcted myocardium [29,30].

GATA-4 is a zinc finger transcription factor that promotes myocardial regeneration [31]. GATA-4 is also considered as an anti-apoptotic factor implicated in regulating cell growth, differentiation, and survival. An inducible loss of GATA-4 severely depressed ventricular function accompanied by reduced CM replication and impaired coronary angiogenesis [32]. The peri-infarct intra-myocardial delivery of GATA-4 prevented adverse post-infarction remodeling through myocardial angiogenesis, anti-apoptosis, and stem cell recruitment [33]. Our previous studies suggest that the transduction of MSCs with GATA-4 (MSC^GATA-4^) significantly promoted MSC-mediated ischemic myocardium salvage, which appeared not only due to increased MSC survival [34] and trans-differentiation [35] but also due to the protection of native CM and the promotion of angiogenesis through the enhanced paracrine effects of MSCs [5]. EVs derived from MSC^GATA−4^ (EV^GATA-4^) are more effective at protecting CM from ischemic injury than EVs from empty vector-transfected MSC (EV^null^) [18]. It is unclear whether EV^GATA−4^ promotes angiogenesis during the regeneration of ischemic myocardium. Given that miRs are encapsulated in MSC-derived EVs and that EVs can be regulated by the modification of their parent cells, we hypothesize that EV^GATA−4^ promotes angiogenesis via delivering pro-angiogenic miRs to recipient cells.

## 2. Materials and Methods

### 2.1. Animals

C57BL/6 mice were purchased from Charles River Laboratories. The experimental protocols are approved by The Animal Care Committee of the University of Cincinnati Institutional Animal Care and Use Committee. All animal experiments were conducted in accordance with the National Institutes of Health guide for the care and use of laboratory animals (NIH publication No. 85-23, Revised 1996). Mice were housed under specific pathogen-free laboratory conditions with optimal temperature, humidity, and photoperiods (12L: 12D).

### 2.2. Generate Stable MSCs with GATA-4 Overexpression

MSC cell line C3H10T1/2 cells were purchased from ATCC (Manassas, VA, USA). MSCs were cultured with Dulbecco’s modified Eagle medium/Ham’s Nutrient Mixture F12 (DMEM/F12)(Hyclone, Logan, UT, USA) supplemented with 10% fetal bovine serum (FBS) at 37 °C in humidified air with 5% CO_2_ atmosphere. Stable MSCs with GATA-4 overexpression were generated using a pMSCV retroviral expression system (Clontech, Mountain View, CA, USA) carrying the GATA-4 gene open reading fragment and the GFP protein gene, and they were selected by puromycin, as we have done previously [18,34]. The control cells (MSC^null^) were transduced using the same retroviral system with the GFP protein gene fragment only.

### 2.3. Isolation and Characterization of EVs

MSCs were first seeded at 3 × 10^6^ per 15-cm plate in complete DMEM/F12 medium for 24 h. Then, the completed medium was replaced with 15 mL of serum-free medium. After 24 h, the culture medium was obtained, and EV were collected using ultracentrifugation method shown as Figure 1. The morphology of EVs was examined under transmission electron microscope (JEOL JEM-1230) (JEOL, Peabody, MA, USA), as described previously [36]. The size and concentration of EVs was evaluated using dynamic light scattering in a particle- and molecular-size analyzer (Zetasizer Nano ZS)(Malvern Instruments, Malvern, UK). The expression of CD9, CD63, HSP70, and calnexin in EVs was quantified using Western blotting. The amount of protein in EVs was determined using the BCA method (Thermo Fisher Scientific, Waltham, MA, USA).

### 2.4. Real-Time PCR

Total RNA was extracted from EVs using mirVana^TM^ miR isolation kit (Ambion, Austin, TX, USA) following manufacturer’s protocol. cDNA synthesis and quantitative PCR were performed using the miScript PCR Starter Kit (Qiagen, Germantown, MD, USA). In brief, 1 μg total RNA obtained from each preparation was used for the first-stand cDNA reverse transcription in the 20 μL system. Real-time PCR was performed with specific primers (Qiagen) using a iQ5 real-time PCR system (Bio-Rad, Hercules, CA, USA), according to the manufacturer’s instructions. Data were normalized to the results obtained with primers specific to β-actin and U6, or in the case of EVs by using the synthetic Spike-In control (Ce-miR-39) as an internal control.

### 2.5. Western Blotting

Proteins were extracted using nuclear (Pierce) (Rockford, IL, USA) (for GATA-4) and total protein extraction reagents (Qiagen) according to the manufacturer’s protocol. The proteins were separated using sodium dodecyl sulfate-polyacrylamide gel electrophoresis (SDS-PAGE) and transferred to 0.45 μm polyvinylidene difluoride membrane (Millipore, Burlington, MA, USA). The membranes were blocked with 5% skim milk in TBST at room temperature for 1 h and incubated with primary antibodies of anti-CD9, anti-calnexin (Abcam, Branford, CT, USA), anti-CD63 (Applied Biological Materials, Richmond, BC, Canada), anti-HSP70 (Cell Signaling, Danvers, MA, USA), anti-GATA-4 (Santa CruZ, Santa Cruz, CA, USA), and anti-THBS1 (Sigma-Aldrich, Saint Louis, MO, USA) at 4 °C overnight. After washing three times with TBST, the membranes were probed with HRP-conjugated secondary antibodies (Cell Signaling), respectively, at room temperature for 1 h and visualized using ECL Plus kit (GE Healthcare, Cincinnati, OH, USA). The expression of a specific protein was normalized with histone H3 and β-actin.

### 2.6. Angiogenesis In Vitro

(1)Tube formation by human umbilical vein endothelial cells (HUVECs) was examined in a 24-well plate coated with Matrigel™ (BD Biosciences, San Jose, CA, USA). HUVECs (3 × 10^4^) were seeded on top of Matrigel™. EVs were added into the medium, and plates were placed into an incubator. Images were taken by a phase-contrast microscopy (Olympus, Center Valley, PA, USA) or an Incucyte Imaging System (Essen, Ann Arbor, MI, USA), and the cumulative tube length of the network structure was measured by randomly selected five microscopic fields using Image J software (National Institutes of Health, Bethesda, MD, USA).(2)HUVECs spheroids were generated as described previously [36]. Briefly, GFP^+^ HUVECs were trypsinized and collected in endothelial cell growth medium (Cell Applications) containing 0.2% carboxymethylcellulose (Sigma). HUVECs (500 cells/100 μL/well) were plated in non-adherent round-bottom 96-well plates (Greiner, Monroe, NC, USA) for 16 h. The spheroids were then harvested and embedded into Matrigel™ basement membrane matrix (BD Bioscience) in endothelial cell serum free defined medium (Cell Applications, San Diego, CA, USA). The cumulative sprout length per spheroid was calculated by measuring from the farthest migrating point to its tangential line position of each sprout using the segmented lines tool in Image J, version 1.53k (National Institute of Mental Health, Bethesda, MD, USA).(3)For the endothelial cell migration test, HUVECs were seeded in 96-well plates at a density of 2 × 10^5^ cells/well. When HUVECs had become a monolayer, scratches were generated in the center of the well using a sterile plastic 200 μL micropipette tip. Images were photographed at 0 and 12 h, and the width of scratch was measured using Image J software.

### 2.7. Angiogenesis In Vivo

The blood vessel formation in the transplanted Matrigel™ plug was assayed in mice, as described previously [23,36]. Briefly, C57BL/6 mice (6~8 weeks old) were anesthetized with ketamine/xylazine (100/10 mg/kg, IP). Matrigel™ (BD Biosciences) (500 μL), mixed with EVs and 15 U of heparin (Sigma), was subcutaneously injected along the abdominal midline. Two weeks later, plugs were excised, embedded in Optimal Cutting Temperature Compound (OCT), and cut into 7 μm sections. The infiltration of endothelial cells into Matrigel™ plugs was determined by immunostaining for CD31. All the CD31 positive cells were quantified as mean pixel density obtained from the image analysis of six random microscopic fields using Image J software. In another cohort of animals, the red blood cells contained inside Matrigel™ plugs were evaluated by determining hemoglobin content (Drabkin’s reagent kit) (Sigma). Briefly, the Matrigel™ plug was homogenized in deionized water and centrifuged. The hemoglobin concentration in the supernatant was determined by the Drabkin assay. The standard curve was generated using Stanbio™ Cyanmethemoglobin Standard (Stanbio Laboratory, Boerne, TX, USA).

### 2.8. Immunofluorescence Staining

Immunofluorescence staining for GATA-4 and CD31 was carried out, as described previously [18]. Briefly, cells or slices of the Matrigel™ plug were fixed with 4% paraformaldehyde for 30 min and then blocked with 10% goat serum containing 0.3% Triton X-100 for 1 h. The primary antibodies of anti-GATA-4 (Santa Cruz) or anti-CD31 (BD Pharmingen™, San Diego, CA, USA) were incubated at 4 °C overnight. Antibody reaction was visualized with dylight 488- or dylight 594-conjugated secondary antibodies (Molecular Probes, Eugene, OR, USA). Nuclei were stained with 4′, 6-diamino-2-phenylindole (DAPI). Fluorescence images were obtained with an Olympus fluorescence microscope (Olympus, Center Valley, PA, USA).

### 2.9. Internalization of EVs and Transfer of Let-7f

EVs derived from MSCs overexpressing CD63-GFP (MSC^CD63-GFP^) were used to track internalization. Lentivirus carrying CD63-GFP fusion protein (System Biosciences, Palo Alto, CA, USA) was used to transfect MSCs and get MSC^CD63-GFP^, according to the manufacturer’s instructions. MSC^CD63-GFP^ was cultured in serum-free DMEM medium (Corning, Corning, NY, USA) for 48 h, and the medium was collected for EV isolation. EVs (EV^CD63-GFP^) derived from MSC^CD63-GFP^ were added to HUVECs and cultured for 24 h. EV internalization was visualized and photographed using an Olympus fluorescence microscope.

To determine the transfer of exosomal miR, pre-miR precursor (mmu-let-7f) (Ambion, Austin, TX, USA) was labeled with Label IT siRNA Tracker Cy3 Kit (Mirusbio, Madison, WI, USA), according to the manufacturer’s instructions. MSC^CD63-GFP^ were transfected with 10 nM of Cy3-labeled pre-let-7f (MSC^CD63-GFP/Cy3-let−7f^) using X-tremeGENE 9 DNA Transfection Reagent (Roche Life Science, Indianapolis, IN, USA) for 24 h. The medium was replaced with fresh serum-free DMEM medium (Corning) and cultured for another 48 h. EVs were collected from the medium of cultured MSC^CD63-GFP/Cy3-let−7f^ (EV^CD63-GFP/Cy3-let−7f^) and added to HUVECs for 24 h. The transfer of Cy3-let-7f was tracked using an Olympus fluorescence microscope.

### 2.10. Gain- and Loss-Function of Let-7f

To generate overexpressing let-7f HUVECs, pre-let-7f-copGFP plasmid or scrambled plasmid (System Biosciences Company, Mountain View, CA, USA) was first mixed with packaging plasmids and co-transfected into 293Ta cells according to the manufacturer’s instructions. Then, HUVECs were infected with the high titer lentiviral particles of let-7f (HUVEC^let−7f^) or the scrambled control (HUVEC^Scrambled^) for 24 h. After infection with the lentiviral particles, HUVECs positive for GFP fluorescence were selected using puromycin. Similarly, knockdown of let-7f in MSC^GATA−4^ was performed by transfecting MSCs with the high titer lentiviral particles of anti-let-7f (MSC^GATA-4 + anti-let−7f^) and the scrambled control (MSC^GATA−4 + Scrambled^) using anti-let-7f expression plasmid pEZX-AM03 with the mCherry red fluorescence reporter gene and its scrambled control plasmid (Genecopoeia, Rockville, MD, USA). MSCs positive for mCherry red fluorescence were selected using hygromycin.

### 2.11. Luciferase Assay

TargetScan was used to find the potential let-7 target sites. This shows that the 3′-UTR of THBS1 contains the conserved let-7 binding site. The miTarget^TM^ dual luciferase reporter vector (pEZX-MT06) containing the full-length 3′-UTR sequence of the THBS1 and the control vector for pEZX-MT06 (mutated vector) were purchased from GeneCopoeia. The HUVECs were seeded in 24-well plates and transfected with THBS1 3′-UTR luciferase reporter vector or the mutated vector pEZX-MT06 (GeneCopoeia) using X-tremeGENE 9 DNA Transfection Reagent (Roche Life Science) according to manufacturer’s protocol. Then, EV^null^ or EV^GATA−4^ were added into the transfected cell and cultured for 48 h. Cells were lysed and collected, and Firefly and Renilla luciferase activities of cell lysate were measured using a Luc-Pair™ Duo-Luciferase Assay Kit (GeneCopoeia).

### 2.12. Statistical Analysis

Data are expressed as mean ± SEM unless otherwise indicated. Statistical analyses were performed by one-way ANOVA or two-tailed Student t-test to measure significant differences between groups. Results were considered statistically significant when *p* < 0.05.

## 3. Results

### 3.1. Characterization of MSC^GATA−4^ and EV^GATA−4^

Both MSC^GATA−4^ and MSC^null^ were GFP positive, and GATA-4 staining was stronger in MSC^GATA−4^ compared to that in MSC^null^ (Figure 2A). Quantitative RT-PCR indicated that the mRNA of GATA-4 was significantly increased in MSC^GATA−4^, compared to that in MSC^null^ (Figure 2B). EVs derived from MSC^GATA−4^ and MSC^null^ were morphologically diverse round-shaped entities under transmission electron microscope (Figure 2C). The average size of EV^GATA−4^ (54.7 ± 3.3 nm) determined by dynamic light scattering was comparable to that of EV^null^ (53.8 ± 2.3 nm) (Figure 2D). There was no significant difference in the expression levels of three specific molecular markers of EVs, CD9, CD63, and HSP70, between EV^null^ and EV^GATA−4^ (Figure 2E). The purification of EVs was determined by Western blot analysis, and no expression of Calnexin (a cell positive marker) was found in EVs (Figure 2F).

### 3.2. EV^GATA−4^ Promotes Angiogenesis

To determine the optimal concentration of EVs and duration, EVs (0~36 μg/mL) and VEGF (18ng/mL) were added into HUVEC culture medium for testing a tube-like structure formation. The plates were placed into the incubator equipped with an Incucyte Imaging System. The images were captured every two hours for 36 h. The tube-like structure formation of HUVECs reached its peak after 16 h. The real-time images of different treatments are shown in Figure 3A. The effect of EVs on promoting tube-like structure formation at 18 μg/mL was similar to the cells treated with VEGF at 18 ng/ml. Therefore, this concentration of EV, i.e., 10 × 10^9^ EV/mL (NTA results: 5.54 × 10^8^ ± 3.25 × 10^7^/µg protein), was used in the following experiments. The cumulative tube length was significantly longer (49.29 ± 2.32 mm/5.375 mm^2^) in HUVECs treated with EV^GATA−4^ than those treated with BSA in the same protein amount (CON, 20.88 ± 2.87 mm/5.375mm^2^), or with EV^null^ (35.51 ± 4.11 mm/5.375 mm^2^) (Figure 3B). Similarly, the sprout length per spheroid in HUVECs treated with EV^GATA−4^ was significantly longer (469.59 ± 94.19 μm/spheroid) than that treated with BSA (85.41 ± 32.04 μm/spheroid), or with EV^null^ (221.46 ± 34.40 μm/spheroid) (Figure 3C).

To further examine whether EV^GATA−4^ promotes angiogenesis in vivo, Matrigel™ plugs containing EVs were transplanted into mice. Matrigel™ plugs contained EV^GATA−4^ were observed to have the deepest red gross appearance after transplantation into mice for 2 weeks (Figure 4A). The hemoglobin content in the plugs containing EV^GATA−4^ (22.08 ± 6.65 μg/mg plug, *n* = 9) was significantly higher than the plugs with BSA control (2.63 ± 1.46 μg/mg plug, *n* = 8), or EV^null^ (11.17 ± 4.30 μg/mg plug, *n* = 9) (Figure 4B). Neovascularization in Matrigel™ plugs was further visualized by immunofluorescence staining specific to CD31. As shown in Figure 4C, the cells positively stained for CD31 in the plugs containing EV^GATA−4^ had significantly more staining than the plugs containing BSA control, or EV^null^.

### 3.3. EV^GATA−4^ Transfer Let-7 miRs to HUVECs

The expression of miRs in EV^GATA−4^ was compared with EV^null^ using the next-generation miRNA sequence. There were 44 miR, out of a total of 358 miR, that were upregulated in EV^GATA−4^. Among these upregulated miR, let-7 family (including miR-3596) were the ones with significantly higher levels expressed in EV^GATA−4^ than in EV^null^ (Table 1).

Quantitative RT-PCR was used to examine the expression of let-7a, let-7d, let-7e, and let-7f in EVs. The expression of these let-7 miRs was upregulated by 2- to 3-fold in EV^GATA−4^, compared to those in EV^null^ (Figure 5A). The expression of let-7a, let-7d, let-7e, and let-7f in HUVECs treated with EV^GATA−4^ was significantly upregulated, compared to those treated with EV^null^ after 16 h (Figure 5B). To demonstrate the internalization of EVs by HUVECs, EVs collected from MSCs overexpressing CD63 with GFP (EV^CD63-GFP^) were added to HUVEC culture. Six hours later, most of HUVECs exhibited green fluorescence and were strongly positive for CD63 by immunofluorescence staining (Figure 5C). Furthermore, we directly tracked the transfer of let-7f by using EVs collected from MSC^CD63-GFP^ that had also been transfected with Cy3-labeled pre-let-7f (EV^CD63-GFP/Cy3-let−7f^). As shown in Figure 5D, HUVECs treated with EV^CD63-GFP/Cy3-let−7f^ for 24 h not only exhibited the signals of CD63-GFP but also showed Cy3-let-7f (red).

### 3.4. Transfer of Let-7f Play a Critical Role in EV^GATA−4^-Mediated Angiogenesis

To downregulate let-7f in EVs, anti-let-7f was transfected into MSC^GATA−4^. Let-7f expression was reduced significantly both in MSC^GATA−4 + anti-let−7f^ and in EVs derived from these cells (EV^GATA−4 + anti-let−7f^), compared to the cells transfected with negative control (MSC^GATA−4 + Scrambled^) and EVs obtained from control cells (EV^GATA−4 + Scrambled^), respectively (Figure 6A). Meanwhile, the expression of let-7f was also significantly reduced in HUVECs treated with EV^GATA−4 + anti-let−7f^, compared to that in HUVECs treated with EV^GATA−4 + Scrambled^ (Figure 6B). The cumulative tube length was significantly decreased in HUVECs treated with EV^GATA−4 + anti-let−7f^, compared to those treated with EV^GATA−4 + Scrambled^ (Figure 6C). In vivo implantation of Matrigel™ plugs in mice for 2 weeks, and vasculature positively stained for CD31 in the plugs containing EV^GATA−4 + anti-let−7f,^ was also significantly less than in plugs with EV^GATA−4 + Scrambled^ (Figure 6D).

The analysis of computational miRs target prediction with microRNA.org and TargetScan revealed that the 3′ UTR of THBS1 contained the conserved let-7 family binding site (Figure 7A). To investigate whether EVs regulated the expression of THBS1 in HUVECs, HUVECs were transfected with miTarget^TM^ dual luciferase reporter vector containing the full-length 3′ UTR sequence of the THBS1 and mutated 3′ UTR of THBS1, respectively. Then, these cells were treated with EV^null^ or EV^GATA−4^ for 48 h. The luciferase activity in HUVECs transfected with the THBS1-3′ UTR luciferase reporter vector was significantly inhibited after being treated with EV^GATA−4^, compared with that in HUVECs treated with EV^null^ (Figure 7B). However, the luciferase activity in HUVECs with mutated THBS1-3′ UTR was not affected under different treatments (Figure 7C). Western blotting results showed that the protein level of THBS1 was significantly reduced in HUVECs treated with EV^GATA−4^, compared with that in HUVECs treated with BSA (as a control) or EV^null^ (Figure 7D). Consistent with this, the protein level of THBS1 was increased by 1.7-fold in HUVECs treated with EV^GATA−4 + anti-let−7f^, compared to that in HUVECs treated with EV^GATA−4 + Scrambled^ (Figure 7E).

Furthermore, the gain of function of let-7f in HUVECs was used to examine the direct effect of let-7 on angiogenesis. HUVECs were directly infected with lentiviral particles of let-7f (HUVEC^let−7f^) or scrambled control (HUVEC^Scrambled^) (Figure 8A). The expression of let-7f was increased by 8.1-fold in HUVEC^let−7f^, compared to that in HUVEC^Scrambled^ (Figure 8B). Western blotting confirmed that let-7f overexpression significantly reduced the THBS1 protein level in HUVECs (Figure 8C). The elevated expression of let-7f was directly associated with the function of endothelial cells. The tube-like structure formation and the migration of HUVEC^let−7f^ (Figure 8D,E) were significantly promoted, compared to that in HUVEC^Scrambled^.

## 4. Discussion

In the present study, the pro-angiogenic effect of EV^GATA−4^ was systematically examined. We found that: (1) EV^GATA−4^ significantly increased angiogenesis, as determined in tests using a tube-like structure formation and spheroid-based sprouting of HUVECs, as well as neovascularization in Matrigel™ plugs implanted in mice; (2) EV^GATA−4^ transferred let-7 miRs down-regulated the expression of THBS1, an anti-angiogenesis protein in target cells. These results demonstrated that the transfer of let-7 miRs from EVs into endothelial cells played a critical role in EV^GATA−4^-mediated angiogenesis.

Neo-angiogenesis is considered a prerequisite for tissue repair and functional recovery associated with stem cell therapy. Several laboratories have reported that EVs derived from MSCs improve cardiac function by increasing capillary density and directly promoting angiogenesis [37,38,39,40,41]. In this study, we systematically determined whether EVs derived from MSCs engineered with GATA-4 were more efficacious in promoting angiogenesis, compared to those derived from MSCs transfected with empty vector. Our results showed that EV^GATA−4^ not only promoted tube-like structure formation and spheroid-based sprouting in vitro but also increased blood vessel formation in Matrigel™ plugs in vivo compared to those derived from control MSCs, which were transfected with an empty vector. Our results are consistent with previous reports that HUVECs can acquire MSC-derived EVs, resulting in an enhancement of in vitro proliferation, migration, and tube formation [39,42].

It is well known that EVs released from stem cells are enriched in miRs [43,44,45,46] and can promote angiogenesis via delivering various pro-angiogenesis miRs [47]. Our previous report [18] indicates that EVs released from GATA-4-engineered stem cells have much potential to protect cardiomyocytes and regenerate ischemic myocardium by releasing multiple miRs responsible for the activation of the cell survival and pro-angiogenesis signaling pathways. We have also demonstrated that exosomal transfer of pro-angiogenic miRs plays an important role in MSC-mediated angiogenesis and stem-cell-to-endothelial-cell communication [36]. In the present studies, we compared the expression of let-7 miR family in EV^GATA−4^ with those in EV^null^. Our results are consistent with a previous report that the expression of these miRs was upregulated in EVs derived from GATA overexpressing MSCs [48].

We selected let-7f, a member of the let-7 miR family, as an example to determine the role of the let-7 related signaling pathway in EV^GATA−4^-mediated angiogenesis using gain-and loss-of-function approaches. We found that increased let-7 miRs played an important role in EV^GATA−4^ mediated angiogenesis. It has been reported that let-7 miRs enhance angiogenesis through protecting microvascular endothelial cells [49]; preventing the dysfunction of endothelial progenitor cells [25]; promoting microvascular pericyte differentiation [24]; and regulating inflammation [50], as well as autophagy and apoptosis [51]. Bioinformatics analysis indicated the putative binding sites of let-7 miRs in the 3’ untranslated regions of THBS1 mRNA. Our studies directly demonstrate that THBS1 is one of the potential targets of let-7 miRs and that the pro-angiogenic effects of let-7 miRs might be associated with the down-regulation of THBS1 expression in endothelial cells. THBS1 is a secreted protein with a variety of biological features, including having a potent anti-angiogenic activity [52]. It inhibits angiogenesis either through direct effecting endothelial cell migration, proliferation, survival, and apoptosis [53], through antagonizing the activity of VEGF [54,55], or possibly by multiple coordinated actions.

In conclusion, our findings revealed that EVs secreted from GATA-4-overexpressing mesenchymal stem cells promoted angiogenesis via delivering let-7 miRs to endothelial cells, resulting in THBS1 downregulation. These results suggest that EV^GATA−4^ could be a viable therapeutic candidate for the induction of therapeutic angiogenesis in wound healing and tissue regeneration, following ischemic events.

## Figures and Tables

**Figure 1 cells-11-01573-f001:**
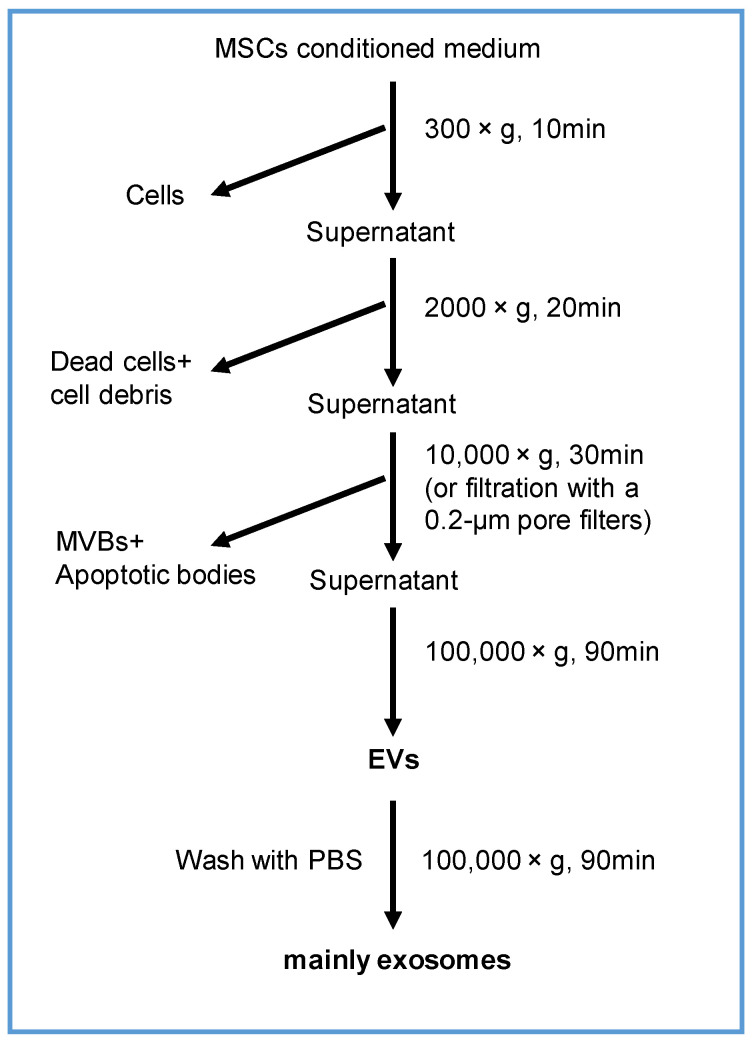
A schematic diagram of the EV collection from cultured MSCs.

**Figure 2 cells-11-01573-f002:**
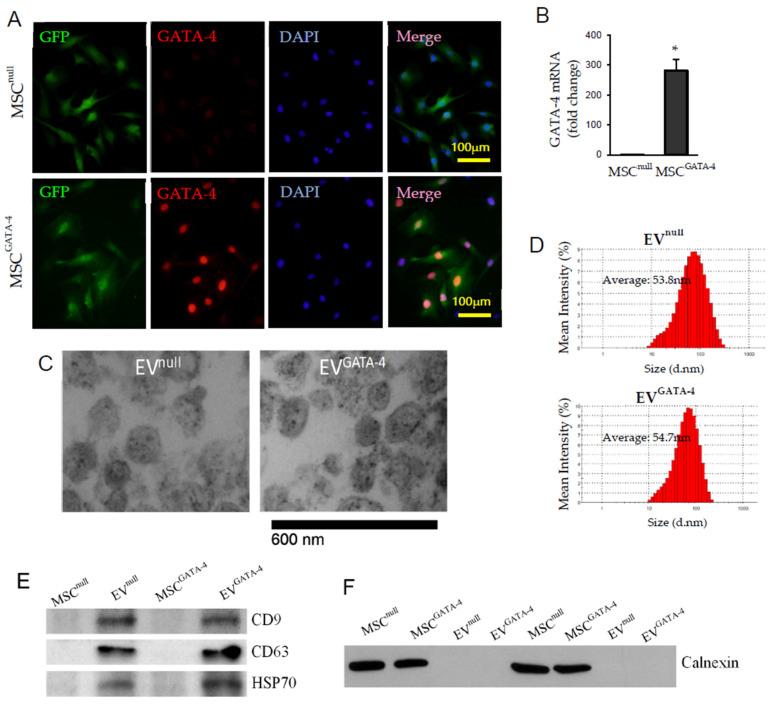
MSC^GATA−4^ construction and EVs identification. (**A**) Immunofluorescence staining showed GATA-4 overexpression in MSC^GATA−4^ nucleus. (**B**) The mRNA of GATA-4 in MSCs determined by qRT-PCR, * *p* < 0.05 vs. MSC^null^. (**C**) The morphology of EVs under transmission electron microscopy. (**D**) The size of EV^null^ and EV^GATA−4^ was measured using a Zetasizer Nano instrument. (**E**) Western blotting of CD9, CD63, and HSP70 in MSCs and EVs. (**F**) Western blotting of Calnexin in MSCs and EVs.

**Figure 3 cells-11-01573-f003:**
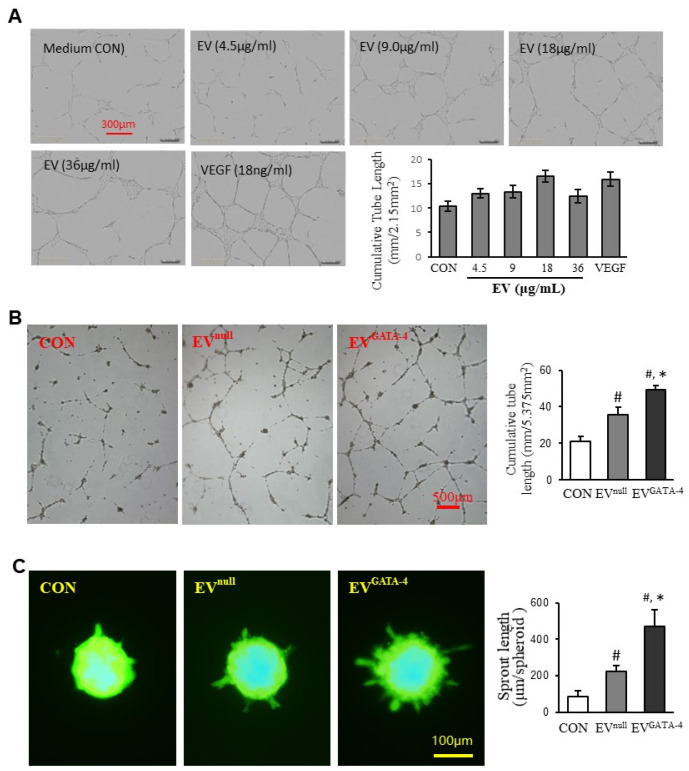
EV^GATA−4^ promoted tube-like structure formation and spheroid sprouting. (**A**) EVs mediated the tube-like structure formation in a concentration-related manner. (**B**) Representative images of capillary-like tube formation and quantitative analysis of the total tube length. (**C**) Representative images of HUVEC spheroids sprouting and quantitative analysis of the cumulative sprout length per spheroid. ^#^
*p* < 0.05 vs. CON (BSA); * *p* < 0.05 vs. EV^null^.

**Figure 4 cells-11-01573-f004:**
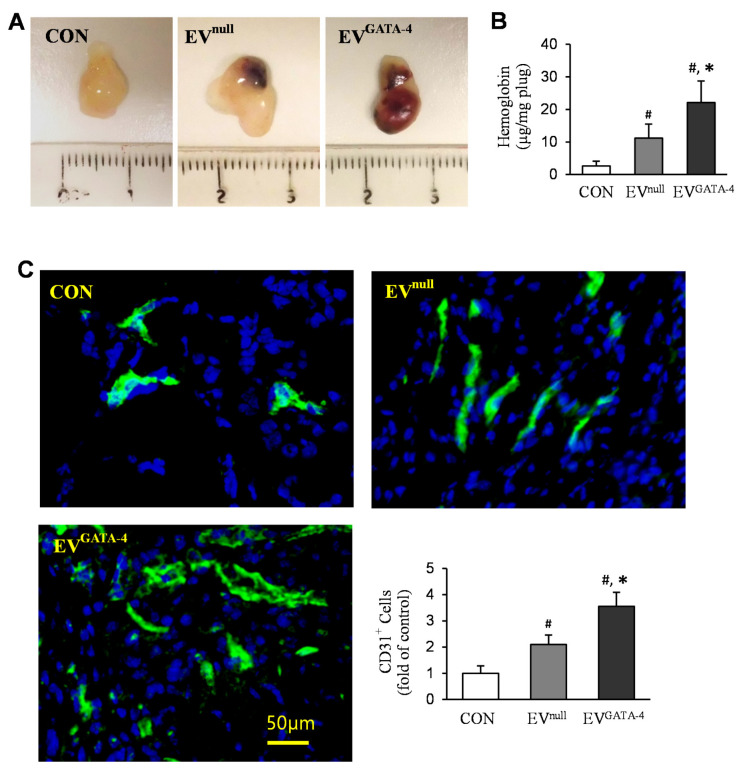
EV^GATA−4^ promoted angiogenesis in vivo. (**A**) The representative gross appearance of Matrigel™ plugs, which were implanted subcutaneously in mice for 14 days. (**B**) Hemoglobin content in the Matrigel™ plugs. (**C**) Immunofluorescence staining of CD31 in the sections of Matrigel™ plugs and quantification of the CD31-positive cells. ^#^
*p* < 0.05 vs. CON (BSA); * *p* < 0.05 vs. EV^null^.

**Figure 5 cells-11-01573-f005:**
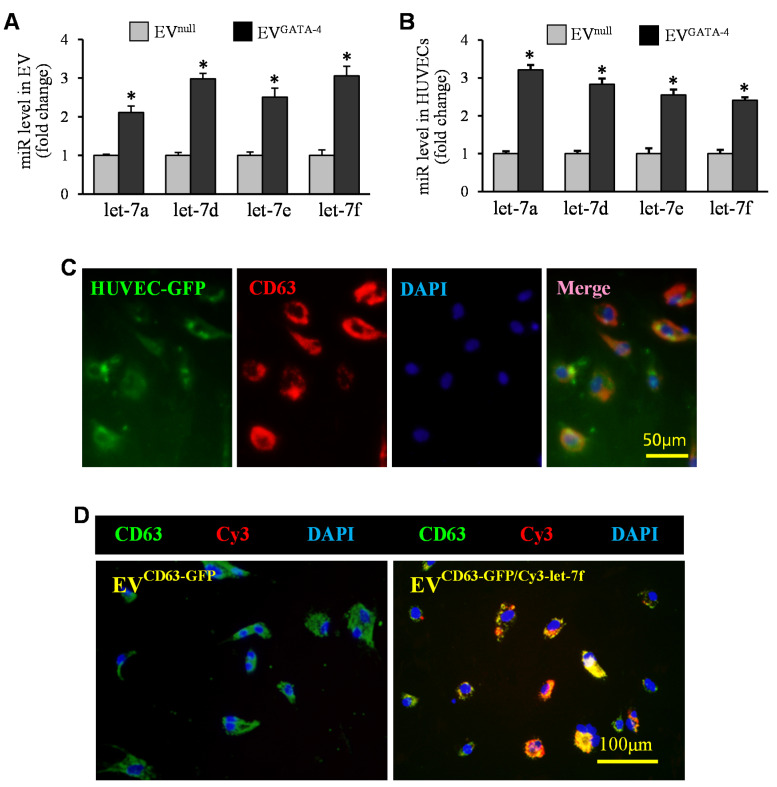
EVs mediated the transfer of let-7 miRs. (**A**) The up-regulation of let-7a, let-7d, let-7e, and let-7f in EV^GATA-4^ was evaluated by qRT-PCR. (**B**) The quantitative analysis of let-7 miRs in HUVECs treated with EV^null^ and EV^GATA-4^. (**C**) The HUVECs showed strong positivity for CD63 after culture with EV^CD63-GFP^. (**D**) Let-7f (red) was strongly expressed in HUVECs cultured with EV^CD63-GFP/CY3-let-7f^. * *p* < 0.05 vs. EV^null^.

**Figure 6 cells-11-01573-f006:**
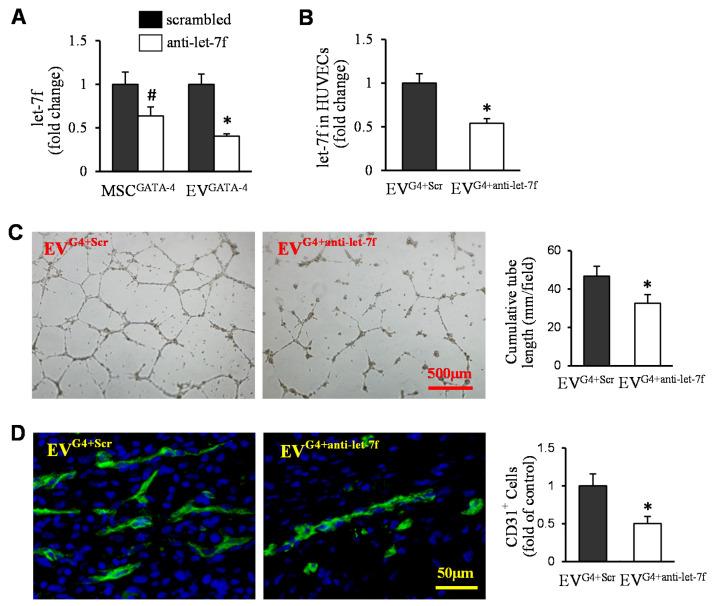
EV^GATA-4^-mediated angiogenesis was related to the expression of let-7. (**A**) The expression of let-7f in MSC^G4 + anti-let-7f^ and EV^G4 + anti-let-7f^, as well as their control. (**B**) The expression of let-7f in HUVECs treated with EV^G4 + anti-let-7f^ and EV^G4-Scr^. (**C**) Representative images of capillary-like tube formation and quantitative analysis of the total tube length following EV treatment. (**D**) Immunofluorescence staining of CD31 in the sections of Matrigel™ plugs, and the quantification of the CD31-positive cells. ^#^
*p* < 0.05 vs. MSC^G4 + Scr^; *****
*p* < 0.05 vs. EV^G4 + Scr^. (EV^G4 + Scr^ = EV^GATA-4 + scrambled^; EV^G4 + anti-let-7f^ = EV^GATA-4 + anti-let-7f^).

**Figure 7 cells-11-01573-f007:**
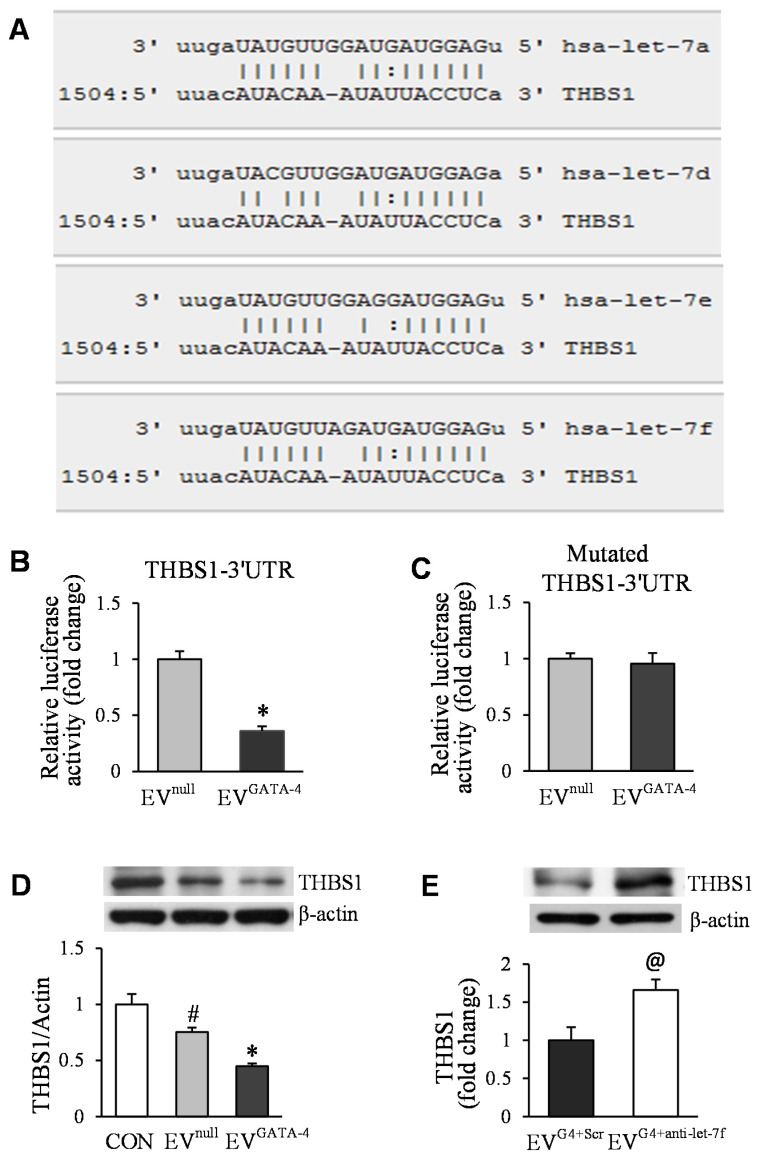
Exosomal let-7 miRs functionally down-regulated the target gene THBS1 in HUVECs. (**A**) An example of complementarity between let-7 miRs and the THBS1 3′UTR. (**B**,**C**) Quantitative data for dual-luciferase reporter assay in HUVECs treated with EV^null^ or EV^GATA-4^. (**D**) The THBS1 protein levels in HUVECs following different treatments determined by Western blotting. (**E**) The protein level of THBS1 in HUVECs treated with EVs. ^#^
*p* < 0.05 vs. CON (BSA); *****
*p* < 0.05 vs. EV^null^; and ^@^
*p* < 0.05 vs. EV^G4-Scr^, respectively. (EV^G4 + Scr^ = EV^GATA-4 + Scrambled^; EV^G4 + anti-let-7f^ = EV^GATA-4 + anti-let-7f^).

**Figure 8 cells-11-01573-f008:**
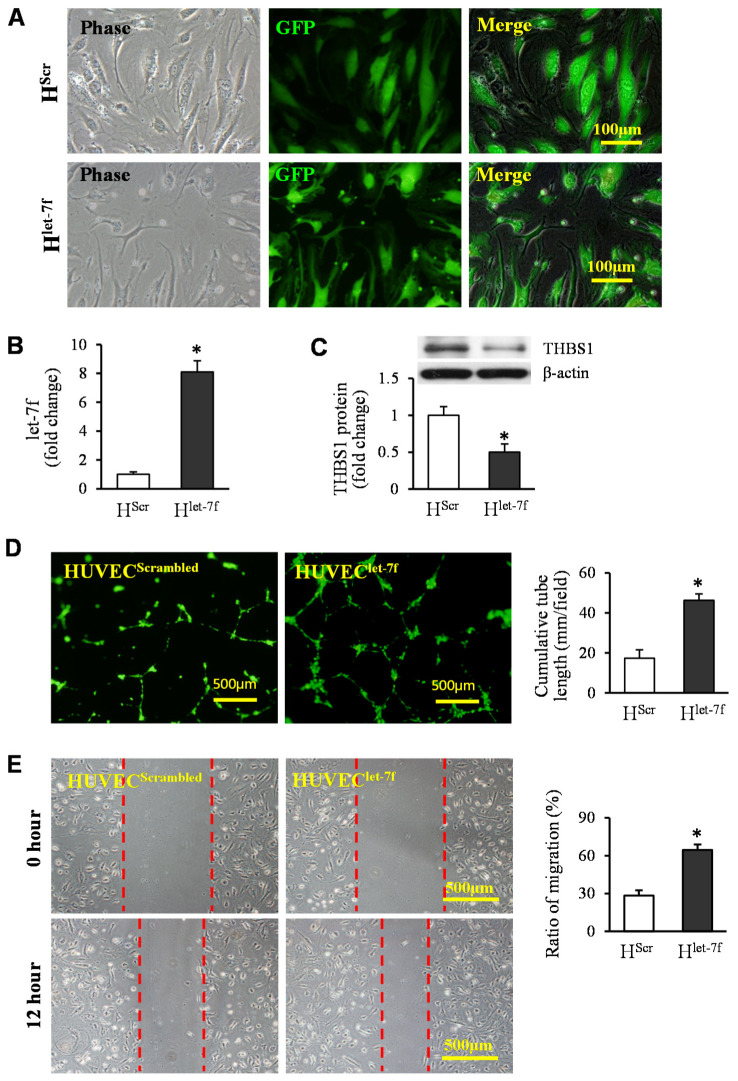
Let-7f overexpression in HUVECs promoted tube-like structure formation and migration. (**A**) Representative images of HUVECs with let-7f overexpression (HUVEC^let-7f^), and their negative control (HUVEC^Scrambled^). (**B**) The expression of let-7f in HUVECs. (**C**) The THBS1 protein in HUVECs transfected with let-7f. (**D**) Representative images of capillary-like tube formation and quantitative analysis of the total tube length. (**E**) Representative images of HUVEC migration and quantitative analysis. * *p* < 0.05 vs. HUVEC^Scr^. (HScr = HUVEC^Scrambled^, H^let-7f^ = HUVEC^let-7f^.)

**Table 1 cells-11-01573-t001:** The expression of let-7 in EVs.

miRs	Reads	
	EV^GATA-4^	EV^null^
miR-3596d	145,686	47,721
let-7f	132,442	43,383
miR-3596a	67,251	11,255
let-7a	66,997	11,385
miR-3596b	60,521	19,136
let-7d	55,019	17,397
let-7f-1	32,765	10,810
miR-3596c	20,236	8314
let-7e	19,317	7936
let-7a-1	15,939	2910
let-7a-2	15,429	2587
let-7f-2	10,254	4072
let-7e	4673	2479

## Data Availability

The datasets generated and analyzed during the present study are available from the corresponding author upon reasonable request.

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
