# Peer review of "Nano-Sized Extracellular Vesicles Secreted from GATA-4 Modified Mesenchymal Stem Cells Promote Angiogenesis by Delivering Let-7 miRNAs"

_cells, 2022, doi:10.3390/cells11091573_

Round 1

Reviewer 1 Report

Gong and group studied the extracellular vesicles secreted from GATA-4 modified mesenchymal stem cells promote angiogenesis by delivering let-7 miRNAs. The study is interesting and well written, but still, the study is required to revise before publication.

I have a few important fundamental comments.

  1. The author failed to prove the purification of EVs, I recommend adding western blots results of GM130 or Calnexin for EVs negative markers (cell positive markers).
  2. The authors during the isolation process, didn’t use filters such as 0.45µm or 0.2 µm, Then how come the EVs size was around 50nm, author should explain?
  3. NTA figure low resolution make hard to see numbers in X and Y-axis.
  4. Morphology of EVs under transmission electron microscopy was shown but I recommend adding transmission electron microscopy imaging of both EVnull and EVGATA-4.
  5. Other experiments were well performed, presented and discussed.

Author Response

We would like to thank you for the insightful and thoughtful comments, which significantly help us to improve the scientific quality of the revised manuscript. Below are our responses to the comments point-by-point.

Reviewer #1: Gong and group studied the extracellular vesicles secreted from GATA-4 modified mesenchymal stem cells promote angiogenesis by delivering let-7 miRNAs. The study is interesting and well written, but still, the study is required to revise before publication.

I have a few important fundamental comments.

1. The author failed to prove the purification of EVs, I recommend adding western blots results of GM130 or Calnexin for EVs negative markers (cell positive markers).

We have done additional western blots for calnexin measurement. The result is presented in Figure 2.

2. The authors during the isolation process, didn’t use filters such as 0.45µm or 0.2 µm, then how come the EVs size was around 50nm, author should explain.

We feel sorry for not explain it well in our original manuscript. Actually, we used the filters (0.2 µm) and centrifuge at 10,000 × g for 30 min to remove MVBs and apoptotic bodies. We have revised “schematic diagram of EV collection from cultured MSCs” (Figure 1).

 3. NTA figure low resolution make hard to see numbers in X and Y-axis.

We are very sorry that the resolution of NTA figures was reduced during these figures were copied and pasted in the previous version. We have replaced these figures with high resolution.

4. Morphology of EVs under transmission electron microscopy was shown but I recommend adding transmission electron microscopy imaging of both EVnull and EVGATA-4.

Thank you for the suggestion. The transmission electron microscopy images of both EVnull and EVGATA-4 have been added.

5. Other experiments were well performed, presented and discussed.

Thank you very much for your positive comments.

Reviewer 2 Report

This manuscript reports that EVs derived GATA-4 overexpressing MSCs show increased pro-angiogenic Activity, associated with let-7 miRs targeting THBS1 in endothelial cells. The work is well described, the methods appear appropriate and the results are fairly reported. This paper can be of interest for the readers of Cells.

Some important quantitative data are missing and should be included before this paper can be accepted. The Authors report that a concentration of EV-protein equivalent to 18 ug/ml was optimal to induce tube formation in vitro. However, they do not report the number of nanoparticles in their preparation, as required in the MISE2018 guidelines. Additionally, the concentration of let-7 miRs in the nanoparticles should be reported, to provide significant information in such a crowded field of research. Indeed, several papers reported biological effects of exosomal miRNAs in different contexts, but their effective concentration is still a matter of debate.

Author Response

We would like to thank you for the insightful and thoughtful comments, which significantly help us to improve the scientific quality of the revised manuscript. Below are our responses to the comments point-by-point.

Reviewer #2: This manuscript reports that EVs derived GATA-4 overexpressing MSCs show increased pro-angiogenic Activity, associated with let-7 miRs targeting THBS1 in endothelial cells. The work is well described, the methods appear appropriate and the results are fairly reported. This paper can be of interest for the readers of Cells.

  1. Some important quantitative data are missing and should be included before this paper can be accepted. The Authors report that a concentration of EV-protein equivalent to 18 ug/ml was optimal to induce tube formation in vitro. However, they do not report the number of nanoparticles in their preparation, as required in the MISE2018 guidelines.

The number of nanoparticles was calculated based on the results of NTA. There are 5.54× 108 ± 3.25 × 107 particles / µg protein. Therefore, the number of nanoparticles we used is about 10 × 109 EV/ml (18 µg/ml), which has been added.

  1. Additionally, the concentration of let-7 miRs in the nanoparticles should be reported, to provide significant information in such a crowded field of research. Indeed, several papers reported biological effects of exosomal miRNAs in different contexts, but their effective concentration is still a matter of debate.

We want to thank the reviewer for the thoughtful suggestion. In this study, there were 44 miR, out of a total of 358 miR, that are upregulated in EVGATA-4 in the results of next generation miRNA Sequence. Among these upregulated miR, let-7 family (including miR-3596) were more significantly with very higher expression level in EVGATA-4. The concentration of let-7mRs in the nanoparticles has been listed in Table 1.

Round 2

Reviewer 2 Report

Accept in present form